# Malnutrition Prevalence in Australian Residential Aged Care Facilities: A Cross-Sectional Study

**DOI:** 10.3390/healthcare12131296

**Published:** 2024-06-28

**Authors:** Marie-Claire O’Shea, Judy Bauer, Clare Barrett, Katina Corones-Watkins, Ursula Kellett, Stephen Maloney, Lauren T. Williams, Christian Osadnik, Jonathan Foo

**Affiliations:** 1School of Health Sciences and Social Work, Griffith University, Gold Coast, QLD 4222, Australia; clare.barrett@griffith.edu.au (C.B.); lauren.williams@griffith.edu.au (L.T.W.); 2Department of Nutrition, Dietetics and Food, Monash University, Clayton, VIC 3800, Australia; judy.bauer@monash.edu; 3School of Nursing and Midwifery, Griffith University, Gold Coast, QLD 4222, Australia; k.corones-watkins@griffith.edu.au (K.C.-W.); u.kellett@griffith.edu.au (U.K.); 4Department of Physiotherapy, Monash University, Frankston, VIC 3199, Australia; stephen.maloney@monash.edu (S.M.); christian.osadnik@monash.edu (C.O.); jon.foo@monash.edu (J.F.)

**Keywords:** long-term care, malnutrition screening, nutrition support, prevalence, dietitians, BMI, weight loss

## Abstract

Long-term or residential services are designed to support older people who experience challenges to their physical and mental health. These services play an important role in the health and well-being of older adults who are more susceptible to problems such as malnutrition. Estimates of the significance of malnutrition require up-to-date prevalence data to inform government strategies and regulation, but these data are not currently available in Australia. The aim of this study was to collect malnutrition prevalence data on a large sample of people living in residential aged care facilities in Australia. A secondary aim was to examine the relationship between malnutrition and anthropometry (body mass index (BMI) and weight loss). This prevalence study utilised baseline data collected as part of a longitudinal study of malnutrition in 10 Residential Aged Care facilities across three states in Australia (New South Wales, South Australia, and Queensland). The malnutrition status of eligible residents was assessed by dietitians and trained student dietitians using the Subjective Global Assessment (SGA) with residents categorised into SGA-A = well nourished, SGA-B = mildly/moderately malnourished, and SGA-C = severely malnourished. Other data were extracted from the electronic record. Of the 833 listed residents, 711 residents were eligible and had sufficient data to be included in the analysis. Residents were predominantly female (63%) with a mean (SD) age of 84 (8.36) years and a mean (SD) BMI of 26.74 (6.59) kg/m^2^. A total of 40% of residents were categorised as malnourished with 34% (n = 241) categorised as SGA-B, and 6% (n = 42) SGA-C. Compared to the SGA, BMI and weight loss categorisation of malnutrition demonstrated low sensitivity and high specificity. These findings provide recent, valid data on malnutrition prevalence and highlight the limitations of current Australian practices that rely on anthropometric measures that under-detect malnutrition. There is an urgent need to implement a feasible aged care resident screening program to address the highly prevalent condition of malnutrition in Australia.

## 1. Introduction

Life expectancy is increasing globally. By the year 2050, it is estimated that the number of people 60 years and older will double, and that those 80 years or older will triple, reaching a global population of 2.1 billion and 426 million, respectively [1]. As people age, their care requirements tend to increase [1]. Long-term care (LTC) or residential aged care (RAC) services are designed to support older people, generally aged 65 years or over [2], who experience significant declines in physical and mental function. These services enable them to maintain a life consistent with their basic rights, fundamental freedoms, and human dignity [3]. The World Health Organization (WHO) estimates that 34% of Americans aged 65 years and older receive care within a residential facility with comparable figures reported in other countries including Australia (35.4%), New Zealand (34.4%), Sweden (34.7%), and Canada (32.4%) [4]. These numbers highlight the importance of RAC services in supporting older adults and ensuring their well-being as they age.

Good nutrition plays a crucial role in healthy ageing. Due to age-related physiological changes, limited access to nutritious food, and the presence of other health conditions, older people are particularly susceptible to malnutrition [5]. According to the European Society for Clinical Nutrition and Metabolism, malnutrition, also known as undernutrition, is characterised by insufficient intake or absorption of nutrients, resulting in altered body composition (reduced fat-free mass and body cell mass) [6]. This condition leads to diminished physical and mental function, frailty, and impaired clinical outcomes from diseases [1]. Malnutrition is associated with poorer overall health, increased utilisation of healthcare services, higher rates of falls and fractures, delayed wound healing, elevated infection risk, and accelerated mortality [7]. Despite such risks, many RAC facilities lack sufficient processes to systematically detect the nutritional status of residents, which places residents at risk of malnutrition [8].

Current estimates of the proportion of RAC residents at risk of or diagnosed with malnutrition vary from less than 29% [5] to over 68% [9]. These figures depend on the specific diagnostic tool used and the country [9,10]. Importantly, many malnutrition prevalence studies have reported small sample sizes and as such, may not accurately represent global, regional, or interfacility trends [11]. When compared to older people living in the community, malnutrition in RAC residents remains a significant societal issue. In a 2021 scoping review, it was found that 31% of 16,190 older people living in the community were either at risk of malnutrition or were malnourished [12]. Given the greater prevalence observed among RAC residents, addressing malnutrition risks in RAC facilities remains a higher priority. Prioritising interventions to combat malnutrition in these settings is crucial for the well-being and quality of life of residents.

For more than 20 years, Australia’s RAC sector, responsible for 193,000 individuals [4], has encountered substantial scrutiny regarding its handling of the malnutrition crisis [13]. In 2021, a national inquiry reported malnutrition rates as high as 68% [9]. However, these estimates were based on a single study with data on just 215 residents collected a decade ago (2013 and 2014) at two RAC facilities in one Australian state [7]. The work of Kellett and colleagues reported malnutrition rates of between 20% and 26% in 101 RAC residents in 2012 [14]. The work by Gaskill in 2008 included 350 residents at eight RAC facilities and reported 6.4% of residents as severely malnourished and 43.1% at risk [8]. While these studies are important, they are now outdated and there is a concerning lack of current prevalence reporting.

With the Australian Government estimating the cost of malnutrition in RAC at approximately AUD 9 billion per annum [11], accurate, timely, and real-time data are essential but not available. The aim of this paper is to provide an updated snapshot of the prevalence of malnutrition in Australian residential aged care facilities.

## 2. Materials and Methods 

### 2.1. Study Design and Participants

This prevalence study relied on baseline data collected as part of a longitudinal study aiming to address malnutrition in 10 RAC facilities across three states in Australia (New South Wales, South Australia, and Queensland). The primary outcome was malnutrition prevalence, with secondary analyses comparing malnutrition classification based on clinical assessment compared to basic anthropometric measures (BMI, weight change). Participating facilities were recruited using convenience sampling from for-profit aged care providers. Facilities were classified according to socio-economic status using the 2021 Index of Relative Socio-economic Advantage and Disadvantage (IRSAD) [15], which assigns a score to each of the 2644 postal areas covering the whole of Australia. The postal area for each facility was identified based on the address, and the IRSAD decile obtained from the Australian Bureau of Statistics, where a higher decile indicates a greater advantage. 

All residents were considered eligible for inclusion unless they were residing in memory-support units or designated dementia units due to the additional challenges associated with obtaining consent and/or collecting data. Excluded residents were those unwilling to participate, not present at the facility/unavailable, or not clinically suitable for involvement. Ethical approval was provided by the Griffith University Human Research Ethics Committee.

### 2.2. Data Collection and Procedures

Data were collected as part of usual care and student training between July and September 2023 by qualified dietitians or dietetics students trained in the collection protocol and supervised by a qualified dietitian (henceforth collectively referred to as clinicians). A paper-based data collection form was developed and completed by clinicians using data from the electronic resident record, speaking with facility staff, and direct resident assessment. 

The malnutrition status of eligible residents was prospectively assessed using the Subjective Global Assessment (SGA) which includes items from a subjective history (e.g., change in dietary intake) and physical examination (e.g., muscle wasting) [16]. The SGA categorises individuals into SGA-A = well nourished, SGA-B = mildly/moderately malnourished, and SGA-C = severely malnourished. The SGA is one of several tools used to assess malnutrition in the aged care setting [17] with a recent study demonstrating the feasibility of students using SGA to complete assessments [18]. Additional variables were retrospectively collected from the electronic resident record to characterise the resident cohort, namely age (in years), gender (male, female), body mass index (kg/m^2^ [BMI]), hospitalisation in the last 6 months (yes or no), diet type (regular, regular easy chew, soft and bite-sized, minced and moist, pureed, or liquidised) [19], and fluid type (thin, slightly thick, mildly thick, liquidised—moderately thick, or pureed—extremely thick), and weight (current and 3-month historical). The data on paper-based forms were entered in a de-identified manner into the cloud-based, purpose-developed database REDCap by a clinician. 

### 2.3. Data Analysis

Data were exported to Microsoft Excel (version 1808) for data cleaning. Records with a missing SGA score were excluded. Data analysis was then conducted in RStudio version 2023.12.1.402 running R version 4.3.2. Descriptive statistics were used to describe malnutrition prevalence and sample characteristics. Associations between resident characteristics and malnutrition status (well-nourished [SGA-A] or malnourished [SGA-B or SGA-C]) were calculated by chi-square test with a significance level of 0.05. Two-by-two matrices were generated comparing malnutrition classification based on SGA with anthropometric cutoffs for malnutrition used in the European Society for Clinical Nutrition and Metabolism guidelines (BMI < 18.5, BMI < 22, and 3-month weight loss > 5%) [20]. Sensitivity and specificity were calculated from the matrices. 

## 3. Results

### 3.1. Participant Characteristics

There were 833 residents listed at the 10 participating RAC facilities. Of these, 39 were ineligible for the following reasons: 33 were residing in memory support units or designated dementia units, 4 were not present at the facility, 1 declined participation, and 1 was deemed clinically inappropriate by the facility clinical nurse. A further 83 records were excluded due to missing SGA scores. A total of 711 residents were included in the final analysis (Figure 1). Residents had a mean (SD) age of 84 (8.36) years, were predominantly female (63%), and had a mean (SD) BMI of 26.74 (6.59) kg/m^2^. The 83 records excluded due to missing SGA scores are likely to be similar to the included participants, with a mean age of 82 years and mean BMI of 26.52 kg/m^2^.

### 3.2. Malnutrition Prevalence

The SGA classifications placed 60% (n = 428) of residents in the well-nourished category, 34% (n = 241) in the mildly/moderately malnourished category, and 6% (n = 42) in the severely malnourished category. Participant demographics stratified according to malnutrition categories are provided in Table 1. There was a statistically significant association between malnutrition status and sex (X^2^ [1, n = 701] = 4.18, *p* = 0.04), hospitalisation within 6 months (X^2^ [1, n = 709] = 13.20, *p* < 0.001), and diet modification (X^2^ [1, n = 708] = 13.31, *p* < 0.001); but not fluid modification (X^2^ [1, n = 707] = 2.49, *p* = 0.11). 

Table 2 provides the SGA scores across all facilities according to area socioeconomic classification. The area socio-economic status of the included facilities ranged from the 2nd (low) to 8th (high) deciles. At the facility level, malnutrition rates (SGA categories B and C combined) ranged from 29% to 50%. There was no apparent pattern between socioeconomic decile and malnutrition rate based on visual inspection. 

### 3.3. Malnutrition and Anthropometry

BMI and weight loss when compared to the SGA demonstrate low sensitivity and high specificity. A BMI with a <18.5 threshold for malnutrition demonstrates 17% sensitivity and 99% specificity; a BMI with a <22 threshold demonstrates 46% sensitivity and 92% specificity; and 3-month weight loss >5% demonstrates 21% sensitivity and 97% specificity (Table 3).

## 4. Discussion

This study presents an updated snapshot of malnutrition status amongst older people living in RAC facilities in Australia. Of the 711 residents assessed across three Australian states, 40% were categorised as malnourished (34% mildly/moderately and 6% severely), supporting the Australian Government’s call for ‘urgent need of improvement’ in the quality and quantity of food provided in residential care [9]. Accurate assessments of malnutrition prevalence are important given the double-edged ‘crisis’ facing society at this time: firstly, a fast growing ageing population [1], and secondly, the RAC sector reported to have “failed to meet the nutritional needs of people for whom they care” [9].

Numerous studies and systematic reviews exist reporting malnutrition rates in hospitalised patients [16,19]; however, the data for residents living in RAC facilities in Australia are limited and outdated [7,14]. While malnutrition rates of between 17.5% and 28.7% are documented internationally, these studies date back over a decade [21]. Some recent European studies indicate prevalence as high as 67.4% [22]. This is much higher than the prevalence observed in the current study. Growing evidence supports the notion that gender is associated with malnutrition risk, with women more likely to be malnourished or at risk of malnutrition than men, and this study supports this notion [23]. Hospitalisations and diet modification show reported links to higher malnutrition rates [22], again in common with the findings in this study.

Regular screening is crucial to identify those who need a full malnutrition assessment and intervention [5]. Unfortunately, systematic surveillance using validated instruments faces several barriers [5], and screening using a validated tool is not routinely performed in Australia. Quality Indicator reporting is mandated by the Australian Government and includes facility staff determining ‘at risk’ residents by applying their own clinical judgement and reasoning using records of BMI and weight loss over a three-month period. Unfortunately, our results demonstrate that while these anthropometric indices have good specificity, their poor sensitivity means many malnourished residents are likely to be undetected and therefore untreated. Thus, to deliver effective care, this sector needs a valid and reliable malnutrition screening instrument that can feasibly be implemented as part of routine care.

We acknowledge that the present study estimates prevalence across 10 facilities in three states and there is no assumption that the rates of malnutrition at these RAC facilities can be generalised across Australia or internationally. The collection of accurate, valid, and reliable national data using standard measures is urgently required to establish the extent of this important problem. A limitation of the present study is that it did not include residents with dementia due to pragmatic consent reasons. Given the recent Australian estimates that 57.4% of residents with dementia are at risk of malnutrition and 27% have a confirmed malnutrition diagnosis [24], this missing population is likely to have resulted in an underestimate of the final prevalence results. Screening mechanisms need to be able to include this important group. A further limitation is the high amount of missing anthropometric data; while this can be adjusted for statistically, it reinforces the problems associated with facilities basing their judgement of malnutrition on these measures.

## 5. Conclusions

This study found that 4 in 10 residents of RAC were malnourished in this sample of 10 centres across three states of Australia. The development of a system that accurately collects up-to-date malnutrition prevalence data is essential. A structured screening or diagnosis program, collated within a national reporting system, is required to hold RAC providers accountable for providing effective care to residents. The next priority for governments and researchers is to develop an evidence-based intervention pathway that enables rapid nutritional response to the life-limiting problem of malnutrition.

## Figures and Tables

**Figure 1 healthcare-12-01296-f001:**
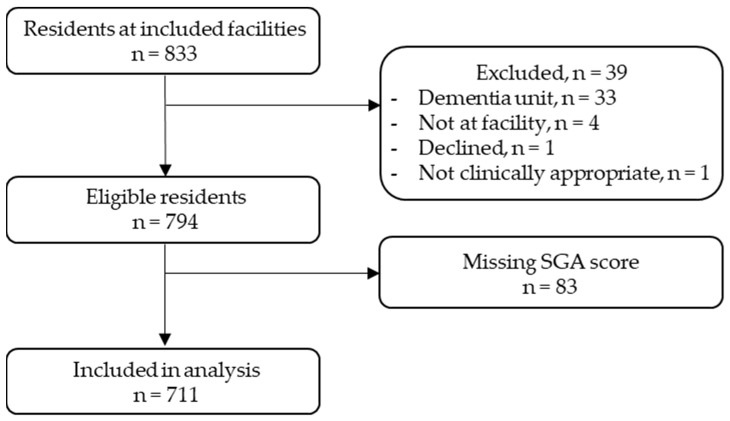
Participant flow diagram.

**Table 1 healthcare-12-01296-t001:** Participant characteristics according to Subjective Global Assessment (SGA) score.

Variable	Well Nourished (SGA-A, n = 428)	Mildly/ Moderately Malnourished (SGA-B, n = 241)	Severely Malnourished (SGA-C, n = 42)	All Participants (n = 711)	Missing Data (n) ^a^
Age, years (SD)	83 (8.51)	87 (7.53)	89 (7.34)	84 (8.36)	5
Gender: Female, n (% ^b^)	277 (63)	137 (31)	28 (6)	442 (100)	10
Male, n (% ^b^)	142 (55)	103 (43)	14 (5)	259 (100)	10
BMI (kg/m^2^), mean (SD)	29.18 (6.60)	23.60 (4.46)	20.11 (4.40)	26.74 (6.59)	148
Hospitalised in last 6 months, n (% ^b^)	106 (50)	91 (43)	15 (7)	212 (100)	2
Modified diet texture, n (% ^b^)	147 (52)	108 (38)	28 (10)	283 (100)	3
Modified fluid type, n (% ^b^)	21 (49)	16 (37)	6 (14)	43 (100)	4

^a^ = Missing data not counted towards totals; ^b^ = percentages calculated from row total.

**Table 2 healthcare-12-01296-t002:** Subjective Global Assessment (SGA) categories by facility.

Facility Number	Socio-Economic Decile ^a^	Age Mean (SD)	Well Nourished (SGA-A)	Mildly/Moderately Malnourished (SGA-B)	Severely Malnourished (SGA-C)	Total Residents
1	8	84 (9.18)	68 (69%)	25 (26%)	5 (5%)	98
2	8	86 (8.48)	36 (61%)	19 (32%)	4 (7%)	59
3	7	85 (8.06)	62 (71%)	22 (25%)	3 (3%)	87
4	7	87 (6.69)	32 (51%)	28 (44%)	3 (5%)	63
5	6	85 (8.17)	35 (54%)	26 (40%)	4 (6%)	65
6	6	82 (8.93)	36 (67%)	16 (30%)	2 (4%)	54
7	4	85 (7.49)	47 (50%)	37 (39%)	10 (11%)	94
8	4	86 (7.27)	32 (54%)	24 (41%)	3 (5%)	59
9	2	81 (9.33)	41 (59%)	26 (37%)	3 (4%)	70
10	2	83 (8.16)	37 (65%)	15 (26%)	5 (9%)	57
Missing		89 (9.02)	2 (40%)	3 (60%)	0 (0%)	5
Total		84 (8.36)	428 (60%)	241 (34%)	42 (6%)	711

^a^ = Based on 2021 Index of Relative Socio-economic Advantage and Disadvantage, where 1 is the most disadvantaged and 10 is the highest advantage.

**Table 3 healthcare-12-01296-t003:** Two-by-two matrix for Subjective Global Assessment (SGA) against anthropomorphic measures for defining malnutrition. For each comparator (BMI or weight loss), the matrix top-left and bottom-right cells indicates concordance with the SGA, whereas top-right and bottom-left indicate non-concordance.

Comparison	Criteria	Malnourished (SGA-B or SGA-C)	Well Nourished (SGA-A)	Total
Body mass index (n = 145 missing)	<18.5 (malnourished)	38	5	43
	≥18.5 (well nourished)	189	334	523
	Total	227	339	566
Body mass index (n = 145 missing)	<22 (malnourished)	104	26	130
	≥22 (well nourished)	123	313	436
	Total	227	339	566
3-month weight loss (n = 360 missing)	>5% (malnourished)	30	7	37
	≤5% (well nourished)	115	199	314
	Total	145	206	351

## Data Availability

The data in this study are available on request from the corresponding author.

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
