# Peer review of "Malnutrition Prevalence in Australian Residential Aged Care Facilities: A Cross-Sectional Study"

_healthcare, 2024, doi:10.3390/healthcare12131296_

Round 1

Reviewer 1 Report

Comments and Suggestions for Authors

Author Response

Pls see responses in attached file.

Reviewer 2 Report

Comments and Suggestions for Authors

Introduction.

I would like to thank you again for the honor of being involved in the review of the article entitled 'Malnutrition prevalence in Australian Residential Aged Care Facilities: a cross-sectional study'. The aim of this study was to collect data on the prevalence of malnutrition in a large sample of people living in residential aged care facilities in Australia.

Malnutrition can occur in aged care residents because of factors such as loss of appetite, eating difficulties, underlying medical problems, dietary restrictions, lack of knowledge and/or resources, and organizational problems.  In the context of an ageing population, the prevalence of malnutrition in aged care facilities in Australia may be of interest. Understanding the prevalence of this problem is essential for assessing compliance with standards of care and identifying areas for improvement. In addition, early treatment of malnutrition may reduce the burden on the health care system by reducing health complications in older residents. In short, in-depth research in this area would help to improve the quality of life of older people and optimize available healthcare resources.

Therefore, I would like to congratulate the authors on this article, which could contribute to a better understanding of ageing and the determinants of health status and life satisfaction in older people.

Our comments will focus on both the form and content of the manuscript, to highlight the improvements that need to be made.

Title: This provides all the information needed to understand the research topic. It includes the type and location of the study as well as the study population.

I. Abstract

This seems to provide all the information expected at this level by briefly recalling the context and mentioning the primary and secondary objectives. The methodology is described. The results are appropriately presented, and a conclusion rounds off this section. We can notice that the Subjective Global Assessment was more sensitive than BMI and weight loss in determining malnutrition.

II. Introduction

We found this section to be well done, presenting all the expected elements (context, rationale, and objective).

III. Materials and methods

We noticed that, although the authors mentioned the non-representativeness of previous studies as a shortcoming, they themselves carried out a study on a convenience sample (lines 92 and 93), which, in our very modest view, does not solve the problem. This limitation, which makes the conclusions non-generalizable, means that it is not possible to conclude with certainty about the prevalence of malnutrition among aged people Residential Aged Care Facilities.

On line 124, the authors mention a version of Microsoft Excel that I am not familiar with. According to very modest research, this version does not appear in the list of Excel versions (1.0 to 2021).

IV. The results

The presentation of the results seems clear to us.

V. Discussion

Further limitations should be added to this section. The results of the study do not seem to be generalizable, although this should have been the ultimate aim of the study.

Author Response

Pls see responses in attached file.

Reviewer 3 Report

Comments and Suggestions for Authors

Comments on the Quality of English Language

Author Response

Pls see responses in attached file.

Reviewer 4 Report

Comments and Suggestions for Authors
  • A brief summary 
  • The aim of the paper was to identify the level of malnutrition in residential care homes in Australia. While there is already research in this area, this is based on a smaller number of participants and there is limited recent research. This study underlines the point that there are likely to be high levels of malnutrition in residential care homes and proposes ongoing screening and a standardised pathway for managing malnutrition   
  • General concept comments
    Article
  • The term LTC and RAC were used interchangeably, perhaps make this clear in the introduction.   

  • Specific comments 
  • Line 41 RAC- identify the differences between LTC and RAC  
  • Line 80 refer specifically to RAC here as this is a key aim of the study
  • Line 114 Justify why you have chosen the SGA over others
  • Lines 118-119- could refer to IDDSI here  https://iddsi.org/ as this will enable others to repeat your study
  • Line 137 -state here is they were likely to be any different from the included participants.
  • Table 2 -to help the reader consider showing the findings by SEC rather than by facility number
  • Discussion and Conclusion Confusion between LTC and RAC LTC and RAC being used interchangeably -clarify in the introduction.
  • Line 170 -it is more than just food in RAC

Author Response

Pls see responses in attached file.

Round 2

Reviewer 3 Report

Comments and Suggestions for Authors

Comments on the Quality of English Language
